# DNA synthesis for true random number generation

Linda C. Meiser[1], Julian Koch[1], Philipp L. Antkowiak [1], Wendelin J. Stark[1], Reinhard Heckel[2] & Robert N. Grass [1✉]

The volume of securely encrypted data transmission required by today's network complexity of people, transactions and interactions increases continuously. To guarantee security of encryption and decryption schemes for exchanging sensitive information, large volumes of true random numbers are required. Here we present a method to exploit the stochastic nature of chemistry by synthesizing DNA strands composed of random nucleotides. We compare three commercial random DNA syntheses giving a measure for robustness and synthesis distribution of nucleotides and show that using DNA for random number generation, we can obtain 7 million GB of randomness from one synthesis run, which can be read out using state-of-the-art sequencing technologies at rates of ca. 300 kB/s. Using the von Neumann algorithm for data compression, we remove bias introduced from human or technological sources and assess randomness using NIST's statistical test suite.

[1] Department of Chemistry and Applied Biosciences, Institute for Chemical and Bioengineering, ETH Zurich, Vladimir-Prelog-Weg 1, CH-8093 Zurich, Switzerland. [2] Department of Electrical and Computer Engineering, Technical University of Munich, Arcisstrasse 21, 80333 Munich, Germany. ✉email: rograss@ethz.ch

"As an instrument for selecting at random, I have found nothing superior to dice. [...] When they are shaken, [...] they tumble wildly about, and their positions at the outset afford no perceptible clue to what they will be after even a single good shake and toss"[1]. These words of Francis Galton published in Nature in 1890, vividly demonstrate one of the simplest methods for generating random numbers. The increasing necessity of being able to generate large quantities of random numbers for societal needs is made obvious when viewing the technological developments thereafter: About half a century later, solving problems with probabilistic procedures demanded a volume of random numbers much greater than that a dice could produce efficiently[2]. Thus began a series of technological breakthroughs including the first integration of a hardware random number generator (RNG) into a real computer, the Manchester Mark I, by using electrical noise[3]. Shifting from algorithm to interactions, the modern world required network security services, and thus introduced encryption and decryption schemes for exchanging information securely, requiring high-quality random numbers (generated faster while being less prone to attacks)[4,5]. New methods for random number generation were developed, such as the Silicon Valley-developed lava lamp and the Mersenne Twister (a software RNG)[6,7]. Of today's state-of-the-art RNGs, the Intel RNG provides 500 MB/s of throughput. Such hardware RNGs create bit streams depending on highly unpredictable physical processes, making them useful for secure data transmission as they are less prone to cryptanalytic attacks[8–11].

It is important to note the distinction between true RNGs and pseudo-RNGs. A true RNG uses a non-deterministic (chaotic) source for random number generation[12,13], whereas a pseudo-RNG creates a deterministic sequence of numbers that depends on an input (seed)[11,12]. If the input seed is known, the entire random number sequence can be reproduced. However, pseudo-RNGs can have better statistical properties and can oftentimes produce random numbers faster than true RNGs, and are thus still popular today. A more recent example of a true RNG has been shown by Gaviria Rojas et al.[14], addressing integrated low-cost, mechanically flexible devices by using semiconducting single-walled carbon nanotubes to digitize thermal noise in order to generate random bits.

As opposed to existing RNGs that are based on physical phenomena or software algorithms, chemical reactions can also be employed as an entropy source for generating random numbers[15]. Chemical reactions are statistical processes where the formation of chemical products follows a certain probability distribution depending on the activation energy for a reaction[16]. Although the expectation of products can be statistically predicted, being able to identify individual molecules after synthesis is rarely possible[15]. Recently, Lee et al. have suggested an automated system exploiting the large available pool of entropy of detectable macrostates of growing crystals in chemical reactions, generating random bits[15]. Although this is a promising approach, not being able to identify individual molecules results in the loss of randomness when analyzing stochastic chemical processes, which is why chemical reactions cannot typically be used as RNGs[15].

This, however, is different for the synthesis of DNA. The synthetic production of DNA is a stochastic chemical process with the advantage that the individual molecules in the synthesized DNA sequence can easily be identified and analyzed by next generation sequencing (NGS) technologies. Sequencing technologies to identify individual nucleotides in strands of DNA have been around since the late 1970s[17]. Nowadays, next-generation sequencing methods offer remarkable throughput[18–20] and enable us to read individual molecules and thus use DNA as a source of random number generation. Previous work has presented the idea of simulation of the DNA random number generation circuitry by theoretically proposing a scheme for a possible automated workflow for DNA random number generation. However, the physical realization of the theory and the experimental limitations were not investigated[21,22].

In this work, we combine the technologies readily available for synthesizing and sequencing DNA to generate random numbers, analyze the results of DNA synthesis, and evaluate the produced randomness. Our contribution is twofold: We offer a transformative application of chemical synthesis as well as explore the robustness and the statistical properties of DNA synthesis.

## Results

**Design of DNA.** In biology, methods for identifying global patterns of the microbial component in the biosphere require the synthesis of random nucleotides at specific positions of primers, to assess for hypervariable regions, for example, of the 16S rRNA gene, to allow for taxonomic classification[23–27]. Other applications for random nucleotide syntheses are found in barcoding, where, by means of unique molecular identifiers (UMI), PCR amplification bias can be eliminated[28]. Such random nucleotides are represented by one single symbol, N, according to the Nomenclature Committee of the International Union of Biochemistry (NC-IUB)[29,30]. Consequently, we have made use of the possibility to synthesize a random nucleotide per position denoted by the letter N in the design of our DNA.

Our DNA strands have been designed such that a 64-nucleotide random region is entailed by a given forward primer region at one end and a given reverse primer region at the other end (see Fig. 1)[27]. The total length of the DNA strand as designed is 105 nucleotides, including the two primer regions and the random region. This DNA strand is then synthesized chemically by suppliers using state of the art solid state synthesis technologies, to obtain a physical medium of randomness (Fig. 2).

The mixing of DNA nucleotide building blocks has also found application in the field of DNA data storage. Anavy et al. have shown that extending the DNA alphabet, by pre-determining the mixing ratio of all four DNA nucleotides at certain positions in the DNA sequence, can increase the logical density for DNA data storage by using composite letters for DNA synthesis[31].

**Analysis of DNA random nucleotide synthesis.** Random DNA sequences as illustrated in Fig. 1 were synthesized commercially three times: twice by Microsynth and once by Eurofins Genomics. We have placed one customized order with Microsynth (synthesis 1), asking specifically for mixing of all building blocks before coupling. The other two orders placed with Microsynth (synthesis 2) and Eurofins Genomics have been ordered regularly on-line, without any special demands. From our DNA order with Microsynth (synthesis 1), we have received 204 μg of dried DNA,

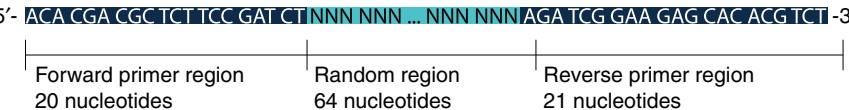

**Fig. 1 DNA design.** Design of DNA containing a forward primer region of 20 nucleotides, random region of 64 nucleotides (where one letter N represents one random nucleotide) and a reverse primer region, containing 21 nucleotides.

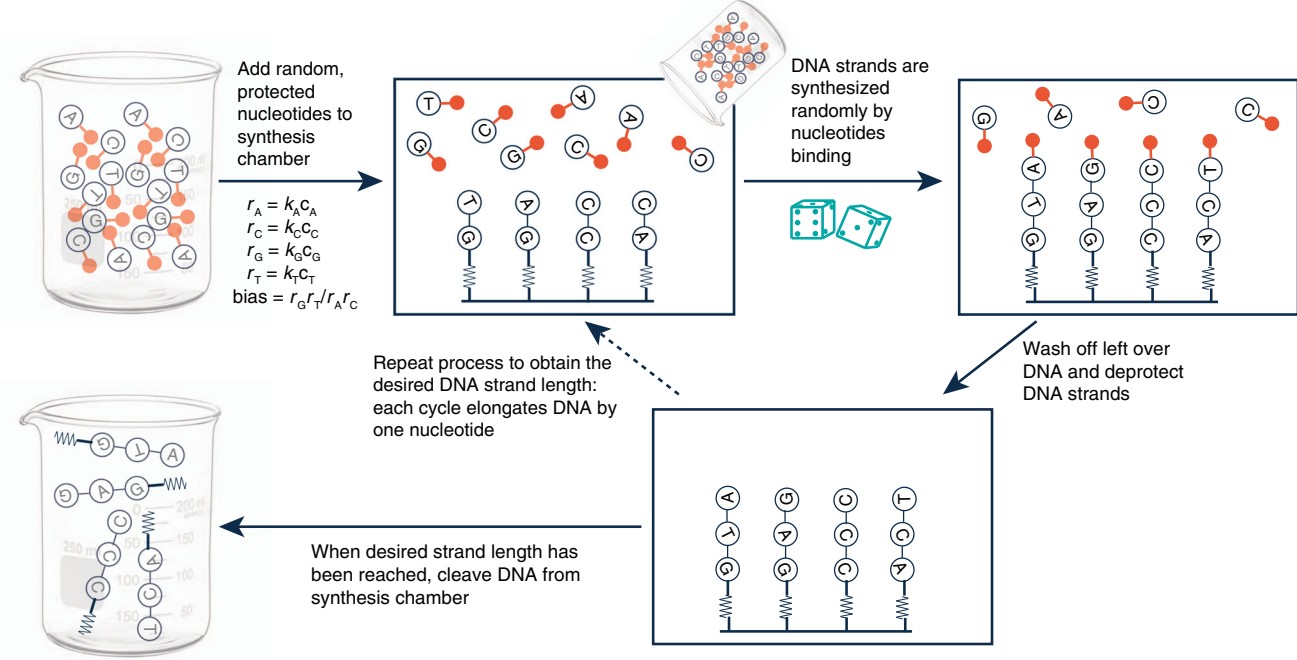

**Fig. 2 DNA random synthesis.** Major procedural steps in synthesizing random DNA strands during solid state DNA synthesis. DNA building blocks are mixed prior to entering the binding substrate, where they start forming a strand of DNA based on their coupling efficiencies. The rate of the individual nucleotides couplings, $r_i$, can be approximated by multiplication of the respective rate constant, $k_i$ and the nucleotide concentration, $c_i$. During the process, individual nucleotides are shielded from binding to other nucleotides using protecting groups, ensuring that only one new random nucleotide can bind per DNA strand per iteration. Excess nucleotides that have not found a DNA strand to bind to are then removed from the synthesis chamber, and DNA strands are de-protected. To elongate each DNA strand to the desired length, the process of adding a mix of nucleotides, washing off left-over and subsequently de-protecting is repeated as often as required. Once the desired strand length of DNA has been reached, the DNA is cleaved from the synthesis support.

synthesized from the 3′ to 5′ direction[32,33]. To read out randomness, the DNA pool was sequenced and subsequently digitally filtered such that sequences not containing the correct adapter were removed (Supplementary Information 1).

When looking at the composition of the DNA strands as a function of position in the random region (Fig. 3) we observe two general trends: (1) In all syntheses, the percentage of G (guanine) and T (thymine) nucleotides is higher than the percentage of A (adenine), and C (cytosine) nucleotides. We call this trend nucleotide nonequivalence. (2) Whereas the percentage of A and C is relatively constant over the string of 60 nucleotides, the percentage of G decreases from 5′ to 3′ and the percentage of T increases from 5′ to 3′. This second trend we call position nonequivalence. This trend is stronger in both synthesis runs from Microsynth than in the material received from Eurofins.

Besides these two general trends, other observations can be made when looking at the synthesis data. Across all manufacturers, the curves in Fig. 3 obtained from Microsynth synthesis 1 are much smoother than those obtained from Microsynth 2 and Eurofins Genomics syntheses.

The observed trends give a first indication about data robustness and could in part be explained by chemical processes occurring during DNA synthesis. The discrepancy between the percentages of nucleotides G, T and A, C (Trend 1, nucleotide nonequivalence) can be caused by several factors. Microsynth informed us in a discussion that the volumes of the individual building blocks are not controlled to the nearest microliter. Differences in concentration across the profile of the mixing chamber may be the result, leading to a less homogeneous distribution of nucleotides along the strand. In addition, the coupling efficiency differs for each building block, and is dependent on variables such as the utilization period of synthesis

reagents by the manufacturers or the protecting groups attached to each building block. The result of differing coupling efficiencies is most likely due to an uneven distribution of the four nucleotides. The decrease of G and increase of T from 5′ to 3′ (Trend 2, position nonequivalence) can be a result of the chemical procedure a DNA strand experiences during synthesis. As DNA synthesis proceeds in the 3′–5′ direction, the nucleotides shown in position 60 of Fig. 3 have been added to the DNA strand first. As synthesized DNA fragments remain in the synthesis chamber until the desired DNA strand length has been obtained, nucleotides added to a DNA strand at the beginning of a synthesis have remained in the synthesis environment for the longest time. Thus, these nucleotides have seen the most synthesis steps, and in turn also the most oxidation steps. This throughput feature of chemical DNA synthesis can be an explanation for trend 2 (position nonequivalence) where the composition of G decreases along the strand in 5′–3′ direction and the composition of T increases in 5′–3′ direction. Oxidation can lead to a phenomenon called G–T transversion, whereby the base G gets chemically altered such that during DNA replication steps, it can be exchanged for a base T. The mechanism of transversion is illustrated in Fig. 3e.

Besides the general trends, the difference in smoothness of the curves in Fig. 3 can be explained by the differences in synthesis strategy from the suppliers: whereas usually the four phosphoramidites are mixed from their storage vessels in the reaction chamber for every synthesis cycle (Micosynth synthesis 2 and Eurofins), the phosphoramidites were pre-mixed (upon request) and this same mixture was used for all synthesis cycles for Microsynth synthesis 1 yielding a much flatter curve.

There are two main potential sources of bias that can have an effect on the results as shown in Fig. 3: coverage bias and error

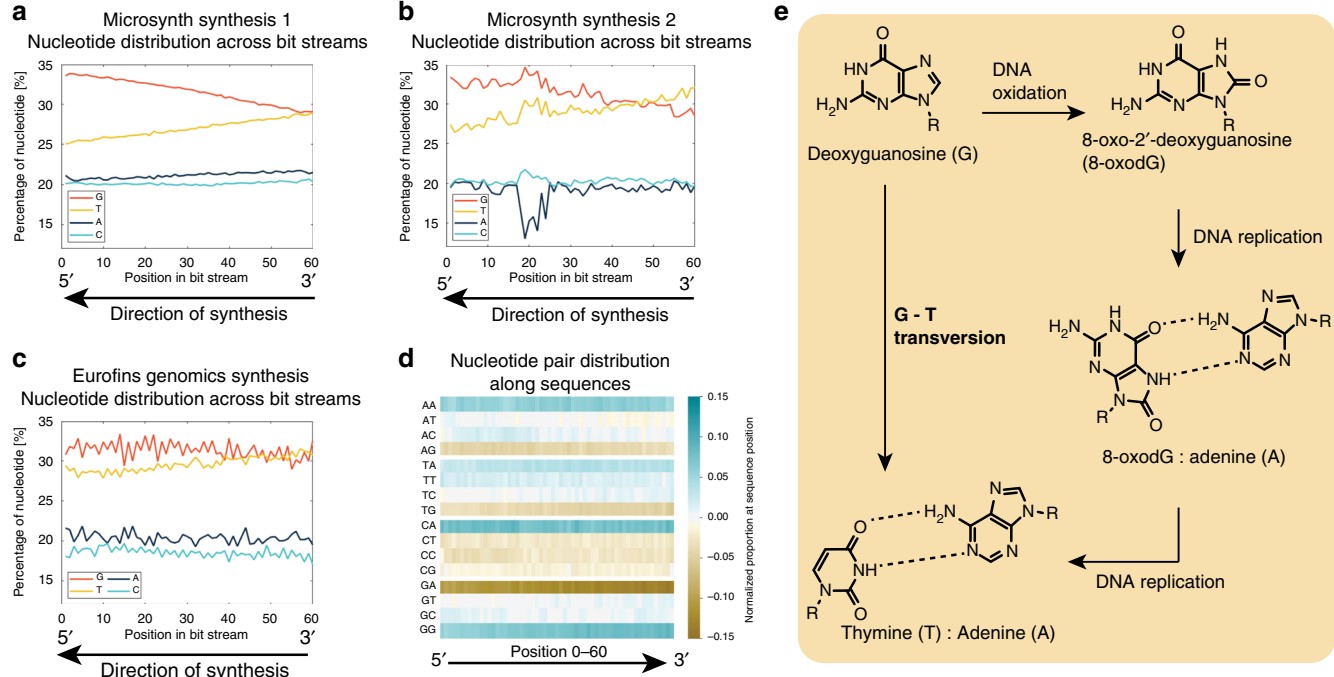

**Fig. 3 Commercial DNA syntheses.** Commercial DNA synthesis of random nucleotides from **a** Microsynth (synthesis 1), **b** Microsynth (synthesis 2), **c** Eurofins Genomics. **d** Heat map showing normalized nucleotide pair distribution along the strand length of Microsynth 1 synthesis (Fig. 3a). For the three syntheses, illustrated are the 60 nucleotides of the random section of DNA strands, respectively, based on the design featured in Fig. 1. The direction of synthesis is indicated by arrows. Analyzed sample size ca. 700,000 sequences each. **e** The mechanism of G–T transversion can be a source for the effect of percentage variation of G and T along the strand of DNA (position nonequivalence). Source data are provided in the Source Data file.

bias. The former bias has been investigated by Chen et al.[34] and is predominantly expressed by bias that can be related to the spatial location on the synthesis chip and PCR stochasticity[34]. The latter bias is the result of insertions, deletions, or substitutions of erroneous nucleotides during synthesis, PCR, and sequencing steps.

For our work, coverage bias only influences the nucleotide distribution if there is a significant discrepancy between coverage of each random sequence. We have analyzed this by counting the number of occurrences of each sequence and found that a single sequence is not present in the pool more than five times with a mean presence of each sequence of 1.03 times. This implies that the bias from sequence coverage cannot be the reason for the observed nucleotide nonequivalence and position nonequivalence behavior.

As for the error bias, it is difficult to distinguish between synthesis and sequencing errors as the two processes cannot be completely decoupled, as access to the molecular morphology of DNA is only possible through sequencing DNA. However, studies have suggested that if data is handled accordingly, sequencing errors occur at random positions[35]. During synthesis of DNA, growing strands may be terminated before having reached the desired length and thus induce a bias to the pool[36,37]. We account for this synthesis bias by following strict sequence selection criteria. To our best knowledge, no studies of sequencing error trends have shown effects of nature and magnitude as we see in Fig. 3a–c. We therefore must conclude that the trend seen is that the trend seen predominately originates from synthesis.

By normalizing Microsynth synthesis 1 (Fig. 3a), we obtain a heat map illustrating the prevalence of two nucleotides binding (Fig. 3d), and can observe a third bias: nucleotide binding prevalence. We see that the preference for one base binding to the existing nucleotide is partially dependent on the nature of the existing nucleotide, thus, guanine is least prevalent to bind to an

adenine (normalized proportion <0), if it has the possibility to bind to an adenine, thymine, cytosine, or guanine, and guanine is most prevalent to bind to guanine (normalized proportion >0), if it is free to bind to adenine, thymine, cytosine, or guanine. Note that the nucleotide pair distribution cannot only be explained by the position-dependent bias as seen in Fig. 3a: For example, when just going by bias, CA should be less frequent than GG at the beginning of the sequence, yet they are equally likely.

From a synthesis point of view, there are practical possibilities to remove the biases induced. For example, adding more T than G building blocks to the reaction as synthesis proceeds (and thus altering the predetermined ratio of nucleotides A, G, T, and C) could leverage the bias from transversion. However, such interventions are laborious and result in the synthesis system to be less robust. We have thus chosen to instead utilize a computational post-processing algorithm to remove bias created during DNA synthesis, increasing the robustness and reproducibility of the overall procedure.

**Data handling.** For data analysis and handling, we treated each synthesis pool shown in Fig. 3 separately. For clarity purposes, we have chosen to demonstrate our results using the data created from Microsynth synthesis 1. Although Microsynth synthesis 1 shows the strongest bias resulting from transversion, the smooth curves show the most homogeneous mixing and coupling behavior during synthesis steps.

Reading the randomness from synthesized DNA strands requires reading of the individual DNA strands, which is done by state-of-the-art sequencing technologies. With various options available for sequencing, we chose Illumina's iSeq100 in a procedure as described by Meiser et al.[38].

Sequencing output (a digital file) was processed in order to select the "error-free" sequences from the pool. Errors that may

have occurred include deletion, insertion, and substitution errors and can result in the DNA strand being short of a base, too long by a base or containing a faulty base, respectively[38]. The selection procedure is illustrated in Supplementary Information 1. Pfeiffer et al.[35] have shown that after processing the DNA pool by means of sequencing-by-synthesis using Illumina, removing (faulty) shortened sequences, errors from sequencing DNA occur at seemingly random positions. Other studies have shown that errors may be related to the sequencing context, where the highest rate of errors was found for nucleotide T, which in our case would result to be erroneous once in every 1250 nucleotides, and the error rates for C, G, and A being much lower[39]. Overall, expected error rates were found to increase towards the end of the DNA strand[39]. To minimize the influence of (especially) deletion errors on randomness, we shortened all sequences to 60 nucleotides and simultaneously selected only the sequences containing the correct length of random nucleotides (Supplementary Information 1).

Once the pool of computer-processed DNA has been confined (to sequences, all 60 nucleotides in length), DNA nucleotides were mapped to bits by using the following scheme: $A \rightarrow 0$, $C \rightarrow 0$, $T \rightarrow 1$, $G \rightarrow 1$. Mapping translated the digitized DNA file to a digitized binary file. (Other mapping alternatives can be found in the Supplementary Information.) The strings of bits (bit streams) obtained after mapping were subsequently tested for randomness using the NIST statistical test suite. We selected very strict randomness evaluation criteria: a pass was only obtained, when all selected tests were passed individually. If even one test failed, the randomness evaluation failed.

Evaluation of the (raw) bit streams using NIST statistical test suite showed that not all tests passed, implying that the bit streams we obtain do not have the same statistical properties as a fully random sequence, i.e., they still contain some redundancy and bias (see Supplementary Information 2). Thus, further bit-processing was necessary in order to remove the bias introduced during synthesis.

**De-biasing algorithm**. There are various options for algorithms removing bias from data. Known are, for example, compression using a cryptographic hash, compression using good linear codes, or compression using the Von Neumann algorithm[9,40,41]. Out of the existing options, we have chosen to apply the Von Neumann corrector to our set of data, as it is very simple to use, and does not introduce any additional source of randomness into the data, allowing the analysis of the random nature of the physical data[9,40]. The output of the Von Neumann corrector is expected to be perfectly unbiased[9,40,41], generating output as follows: (1) if the input is "01" or "10", the first digit becomes the output and the second digit is discarded. (2) If the input is "00" or "11", there is no output and both input digits are discarded. However, the big drawback of the Von Neumann generator is the large loss of data, as at least 75% of the input is being discarded. This implies that the input needs to be sufficiently large[9,40].

The effect of Von Neumann de-biasing can be seen when analyzing the difference between raw bit streams (containing bias) and processed bit streams (stripped of bias). Figure 4 illustrates how Von Neumann de-biasing alters the morphology of the pool of raw bit streams. As Von Neumann de-biasing removes bits from the original bit streams, processed bit streams are shorter than raw bit streams. The cumulative sum of each raw bit stream (each 60 nucleotides long) and each processed bit stream (each shorter than 60 nucleotides long) was calculated by assigning every 0 to the value $-1$ and every 1 to the value 1, and the results were plotted. Further, all de-biased bit streams were put together into one block of bits (bit block).

It was observed that the cumulative sum before de-biasing was skewed (binomial distribution not centered around a cumulative sum across of zero). Removing the bias by applying the Von Neumann algorithm shows a shift of the binomial distribution along the horizontal axis, such that when de-biased, the cumulative sum of the bit distribution is centered around zero. The effect of de-biasing on the bits can also be quantified as follows: Synthesis 1 by Microsynth with a nucleotide-to-bit mapping $A \rightarrow 0$, $C \rightarrow 0$, $T \rightarrow 1$, $G \rightarrow 1$, results in a de-biasing efficiency of 23.7% (meaning 23.7% of bits originally present in raw bit streams are still present after Von Neumann de-biasing). Although the loss of data is massive (more than 75% of all bits lost) and computational efficiency is low (as the average output rate of data is four times slower than the average input rate of data), bias removal is perfect (with the output being completely unbiased, as seen when comparing the cumulative sum across bit streams before and after de-biasing in Fig. 4)[40].

After employing the Von Neumann algorithm, the bit block was tested for randomness again, using the NIST statistical test suite.

**Randomness evaluation (NIST-statistical test suite)**. As can be seen in Table 1, the processed bit streams passed every test listed with a pass rate for every test of >54/56, surpassing the statistically required minimum (52/56)[12]. The decision level for P-values is such that P-value $\geq 0.001$ indicates that the sequence is random with a confidence of 99.9%[12]. The results go to show that the design for the DNA RNG, using the intrinsically stochastic processes of a chemical reaction, is very suitable to serve as an effective true RNG. Robustness evaluation of the NIST statistical test suite with respect to our data is shown in Supplementary Information 3.

As mentioned previously, we have further synthesized two more random pools of DNA (Microsynth, synthesis 2 and Eurofins Genomics synthesis). De-biasing efficiencies for each synthesis scheme are between 23% and 24% (see Supplementary Information 4). Further improvements of de-biasing efficiencies can theoretically be addressed as follows: If we bundle three nucleotides together (for example when taking the nucleotides A, C, and T), we obtain six choices of combination of these three nucleotides, which have the same probability of appearing in a random nucleotide stream. (In our example case, the six possibilities would be: ACT, ATC, CAT, CTA, TAC, TCA.) Each of these possibilities can then be mapped to numbers zero to five, which in turn are mapped to a binary sequence. This binary will be uniformly distributed. If all nucleotides were unbiased, the expected efficiency of de-biasing is 28.72%, thus showing a slight improvement over 1-bit-to-1-nucleotide mapping.

In addition, to the de-biasing efficiencies, results of the NIST evaluation of all syntheses can be found in Supplementary Information 5 with respective analyses and different nucleotide-to-bit mapping schemes applied. It was seen that for every 1-bit-to-1-nucleotide mapping possible, the data set synthesized can be de-biased and made random using Von Neumann algorithms. Additionally, a 2-bits-to-1-nucleotide mapping was investigated. The raw bit streams were not random, and one Von Neumann compression did not make the bit streams random. However, after a second compression, bit streams did become random. These results show the robustness of random number generation using DNA synthesis together with de-biasing using the Von Neumann algorithm.

**Scalability**. The complete process of randomness generation from DNA synthesis is schematically depicted in Fig. 5 where the steps discussed in this work are visually summarized. From the

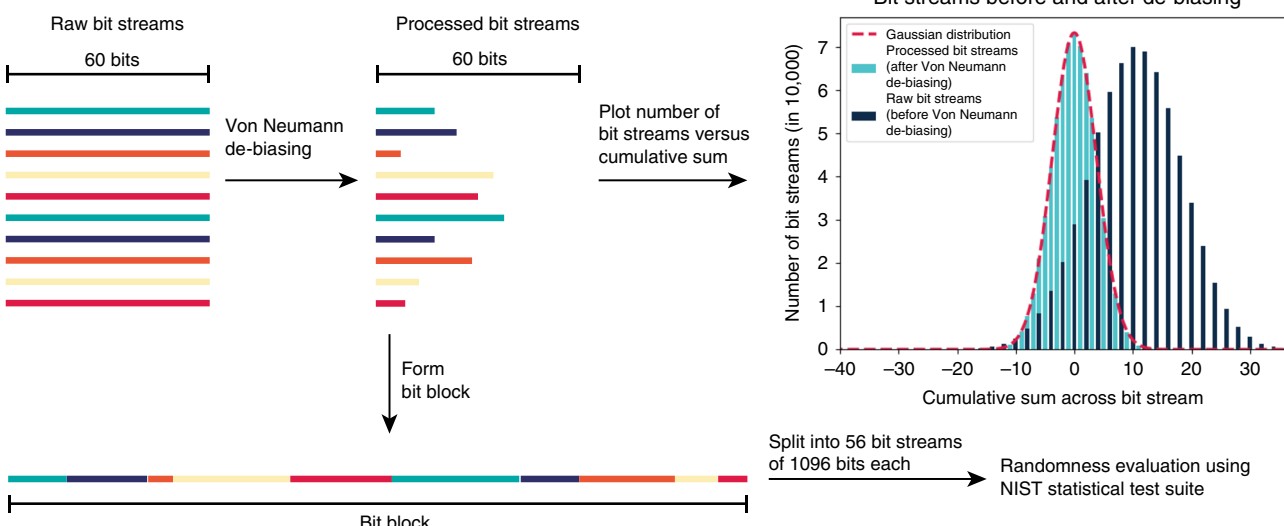

**Fig. 4 Von Neumann de-biasing.** Morphology of pools containing bit streams before and after processing using Von Neumann algorithm (data from synthesis 1 by Microsynth, nucleotide-to-bit mapping A → 0, C → 0, T → 1, G → 1). The cumulative sum across each bit stream in both bit pools was computed by assigning each 0 to the value of −1 and each 1 to the value of 1. The distribution of raw bit streams (dark blue) is not centered around zero, whereas the distribution of raw de-biased bit streams (turquoise) is centered around zero. The expected Gaussian distribution is marked in red. Source data are provided in the Source Data file.

**Table 1 NIST statistical test suite randomness evaluation.**

| NIST statistical test | P-value | Proportion | Result |
|---|---|---|---|
| Frequency | 0.058984 | 56/56 | Pass |
| Block frequency | 0.383827 | 56/56 | Pass |
| Cumulative sums | 0.739918 | 56/56 | Pass |
| Runs | 0.213309 | 55/56 | Pass |
| Longest run of ones | 0.23681 | 54/56 | Pass |
| Rank[a] | 0.616305 | 56/56 | Pass |
| Discrete Fourier transform | 0.002758 | 55/56 | Pass |
| Approximate entropy | 0.574903 | 56/56 | Pass |
| Serial | 0.011791 | 56/56 | Pass |

NIST statistical tests performed on processed synthesized DNA oligonucleotides, mapped to bits using the scheme A → 0, C → 0, T → 1, G → 1. For each test, 56 bit streams containing 1096 bits were tested.
[a]Exception for the rank test: 56 bit streams containing 100,000 bits each were tested.

synthesis by Microsynth (synthesis 1), we have obtained 204 µg of DNA, which translates to about $4 \times 10^{15}$ strands of DNA. Microsynth informed us that this amount of DNA is manufactured by fully automated machines within 8.75 h, and can be obtained commercially for a price of ca. USD 100. The sample of dry DNA contains a theoretical entropy of 28 PB (if there is no bias in the data), and 7 PB of randomness when removing bias using Von Neumann with a loss of 75% of bits. Consequently, in contrast to DNA data storage[38], DNA synthesis is not the bottleneck in DNA random number generation as such a standard synthesis can generate true randomness at a rate of 225 GB/s at a cost of 0.000014 USD/GB.

Sequencing, however, is one of the bottleneck steps concerning time and costs of DNA handling[42]. While our smaller scaled sequencing experiments were performed on an Illumina iSeq device, Illumina technology is fully scalable and high throughput systems (such as Illumina NovaSeq 6000) allow a throughput of up to 20 billion sequence reads within 36 h[43] for a price of ~USD 22,000[44]. Accounting for losses due to strict sequence filtering, and after Von Neumann correction, a full NovaSeq 6000 run could generate true random numbers at a rate of ca. 300 kB/s at a

cost of 600 USD/GB. A more modular approach will allow the combination of different synthesis and sequencing methods, thus, technological advances can further reduce costs of writing and reading randomness from DNA.

In comparison to other random number generation methods (Table 2), DNA synthesis shows a higher randomness production rate than many commercial options such as the online distributors Random.org or HotBits, for example[45,46]. However, randomness generation can also be orders of magnitudes faster, especially when shifting from true RNGs to pseudo-RNGs. When using random numbers transmitted through the internet it is important to consider that these come with limitations, such as potential interception of data, which adds a certain degree of uncertainty to the security.

Limitations in terms of costs and speed of random number generation by DNA synthesis, especially of reading the DNA, will not vanish within the next months. However, there are other benefits entailed to generating randomness in the form of DNA. These include enhanced storage capacities, as the density of data in DNA is very high, mobility of sequencing devices, such as the MinION to a wide range of surrounding conditions, as well as the advantage that once synthesized, DNA can be archived for millennia, preserving the generated randomness over generations to come[47,48].

## Discussion

In this work, we have found that DNA syntheses have two major biases in part from synthesis discrepancies, such as mixing and coupling efficiencies, but also from transversion, originating from oxidation steps during DNA synthesis. These biases can, however, be removed using randomness extraction algorithms such as the Von Neumann algorithm applied here. Using NIST's statistical test suite for evaluation of randomness, we find that de-biased sets of DNA-generated bit streams are random, with all tests passed.

We have shown that DNA synthesis can be used for generating random numbers by making use of the stochastic process of chemical product formation together with next-generation sequencing technologies to analyze individual molecules. By

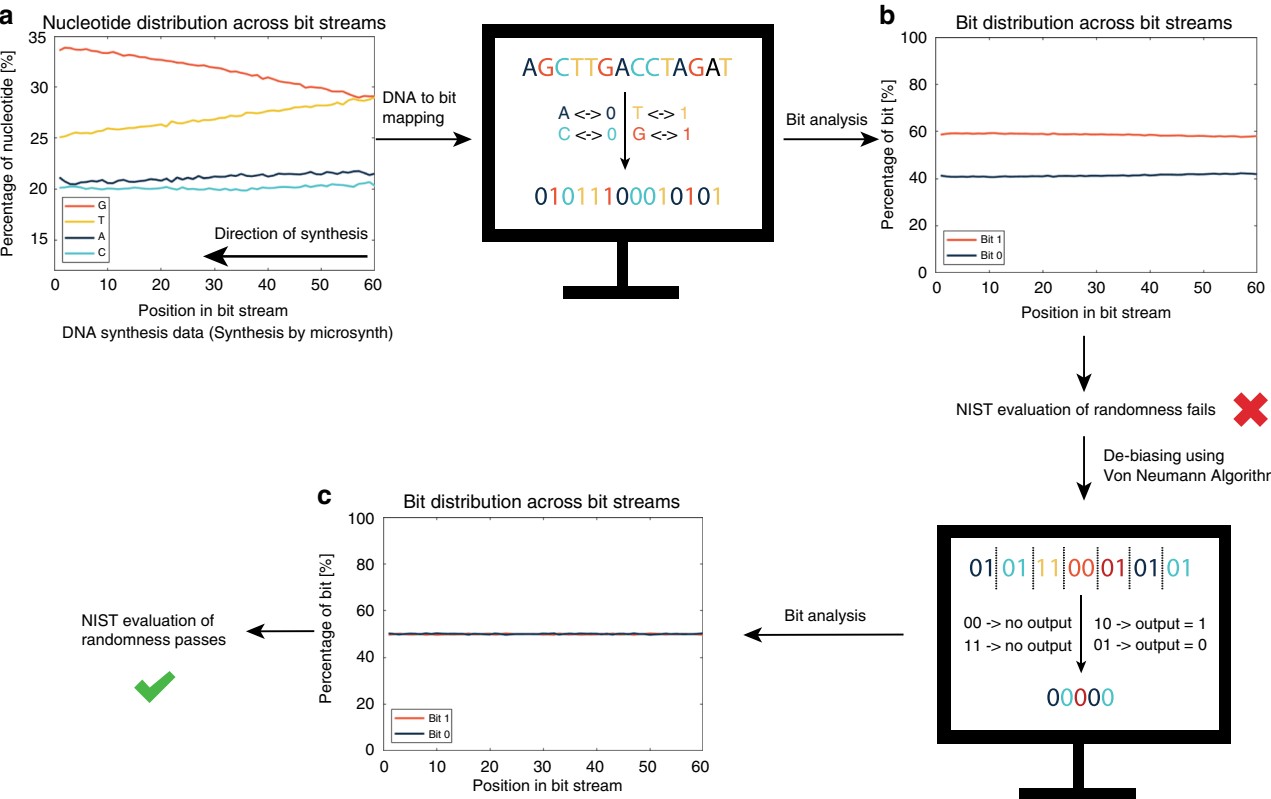

**Fig. 5 Process overview.** Data processing steps for DNA random number generation. **a** DNA synthesis by Microsynth (synthesis 1) showing the percentage of nucleotides per position in the DNA segment for the overall pool (see Fig. 3). **b** Percentage of bits at each position in the bit stream for all sequences in the pool. **c** Percentage of bits at each position in the bit stream for all sequences in the pool after Von Neumann de-biasing algorithm has been applied. Subsequently to Von Neumann de-biasing, the individual bit streams were combined into a block of bits and then separated into 60-bit streams for better comparison (Supplementary Information 6). Source data are provided in the Source Data file.

**Table 2 Selection of random number generators, the underlying generation methods and randomness production rate in MB/s.**

| Random number generator | Randomness production rate [MB/s] | Method |
|---|---|---|
| Meiser et al. | 0.3 | DNA synthesis |
| Gaviria Rojas et al.[14] | Not available | Solution-processed carbon nanotubes |
| Lee et al.[15] | 0.025 | Crystallization robot analyzing chemical processes |
| Reidler et al.[49] | 1560 | Chaotic semiconductor laser |
| HotBits[46] | 0.0001 | Timing successive pairs of radioactive decays |
| Random.org[45] | 0.0015 | Entropy from atmospheric noise |
| Lavarand[6] | 0.02 | Patterns photographed off floating material in lava lamps |
| Intel digital random number generator[8] | 800 | Processor resident entropy source to seed hardware-implemented entropy from atmospheric noise |
| Mersenne Twister[7] | 15,000 | Pseudo-random number generator: algorithm using polynomial algebra |

synthesizing 204 μg of DNA, we have shown the possibility of synthesizing random numbers at a rate higher than 225 GB/s, offering volumes of up to 7 million GB of randomness for a cost of 0.000014 USD/GB (synthesis) and a fully scalable read-out on demand using Illumina sequencing technology. Our analysis further displays that synthesis errors (gradients and nucleotide nonequivalence) commonly observed during the generation of random nucleotides can be computationally corrected for using a standard de-biasing routine. In the examples used the yield of random numbers per strand sequenced was hardly affected by the quality of the synthesis.

DNA as a physical medium containing randomness, can be stored and preserved for millennia to come. Due to the high-randomness density of DNA, it can physically or digitally be transported to any location desired. In this work, we have taken advantage of the stochastic properties of chemical reactions, generating true random numbers from DNA synthesis, offering a viable alternative for large volumes of randomness. While efforts are ongoing, reducing costs for reading and writing DNA, utilizing DNA as a commercial RNG could already be of interest today.

## Methods

**Random-DNA design**. DNA was designed such that two priming regions around 64 random nucleotides give a total DNA length of 105 nucleotides with a structure

as follows: 5′ ACA CGA CGC TCT TCC GAT CT–RANDOM–AGA TCG GAA GAG CAC ACG TCT 3′.

**Random-DNA synthesis.** Three orders were placed with two different DNA synthesizing companies: Microsynth (synthesis 1 and 2), and Eurofins Genomics (synthesis 3). For synthesis 1, we had placed a special demand that all building blocks were to be mixed before coupling. Synthesis 2 and 3 were regular on-line orders (no 5′ and 3′ modifications, 0.2 μmol synthesis scale, PAGE purification).

**Library preparation.** All samples were diluted in 200 μL mili-Q water. Sequencing adapters F1/R1 and 2RI/2FU were added, and dilutions of the samples for sequencing were performed using the protocol described by Meiser et al.[38]. Concentrations were measured using a fluorescence-based DNA concentration assay (Qubit). A list of all primers used can be found in Supplementary Information 7.

**Next-generation sequencing.** Each sample was diluted to 1 nM and then further processed for sequencing using the iSeq100 Sequencing System Guide. For quality control, 2% (vol/vol) PhiX were added to the sequencing run. (PhiX is a reliable, adapter-ligated, ready-to-use genomic DNA sequencing control library provided by Illumina.) Sequencing was performed on Illumina's iSeq 100 with a paired-end read length of $2 \times 150$ bp. Analysis of sequencing output was performed using Matlab version R2018b and python version 3.8.

**Selection of error-free sequences.** To only allow for sequences that have been synthesized and sequenced correctly, sequences containing 16 nucleotides of the adapter (AGA TCG GAA GAG CAC A) were searched for and used for further data treatment. All other sequences were discarded. The remaining sequences were then shortened to 69 nucleotides. As deletions may occur during synthesis and sequencing steps, the designed random region of 64 nucleotides may be shorter than designed. Thus, the remaining 69-nucleotide long sequences were searched for the first 9 nucleotides of the adapter (AGA TCG GAA). This time, all sequences still containing these nucleotides were discarded from the pool. All other sequences were shortened to 60 nucleotides. Thus, it is guaranteed, that the random region does not contain any adapter nucleotides due to deletion errors. An illustration of this procedure can be found in Supplementary Information 1.

**Nucleotide-binding prevalence.** For every position × along the sequence data, the relative presence of two nucleotides following each other (e.g. AT) is normalized by the relative presence of the single nucleotides (e.g. A and T) at the corresponding position. (e.g. for AT at position $x$: $f_x^{\text{norm}}(\text{AT}) = ((f_x(\text{AT})/f_x(\text{A}))/(f_{x+1}(\text{T})))$. This gives the general formula with nucleotides $N_1$ and $N_2$: $f_x^{\text{norm}}(N_1N_2) = ((f_x(N_1N_2)/f_x(N_1))/(f_{x+1}(N_2)))$.

**NIST evaluation parameters.** NIST evaluation tests were chosen such that for each test, 56 bit streams containing 1096 bits were tested (with the exception of the rank test, where 56 bit streams containing 100,000 bits each were tested). This variant of statistical analyses was chosen and explained by Rojas et al.[14]. The tests were applied with the standard parameters as given by the NIST statistical test suite[12], with the following parameters differing from set values: (1) frequency test, no parameter adjustments; (2) block frequency test, no parameter adjustments (block length, $M = 128$); (3) cumulative sums test, no parameter adjustments; (4) runs test, no parameter adjustments; (5) longest runs of ones test, no parameter adjustments; (6) rank test, no parameter adjustments; (7) discrete Fourier transform test, no parameter adjustments; (8) approximate entropy test parameter adjustment: block length, $m = 5$; 9) serial test no parameter adjustments (block length, $m = 16$). For each statistical test, the NIST software computes a $P$-value, which gives the probability that the sequence tested is more random than the sequence a perfect RNG would have produced. Thereby, a $P$-value of 1 indicates perfect randomness, whereas a $P$-value of 0 indicated complete non-randomness. More specifically, for a $P$-value $\geq 0.001$: sequence can be considered random with a confidence of 99.9%, and for a $P$-value $< 0.001$ sequence can be considered non-random with a confidence of 99.9%[12].

**Calculations of cost and rate for randomness generation.** Cost and rate of randomness generation were calculated as a basis of the number of random nucleotides synthesized. This was done by calculating the number of strands synthesized (NS) from the amount of DNA synthesized ($M$): NS = $M \times$ Avogadro constant.

For the number of random bits synthesized (NRB), 60 random nucleotides per strand were assumed. The randomness output volume (ROV) was calculated by assuming a 25% de-biasing efficiency: ROV = $0.25 \times$ NRB.

The possible entropy of a random 60-mer is $4^{60} = 10^{36}$, and thus is significantly larger than the number of sequences in the pool. This implies that it may be expected that every strand synthesized is unique. As a result, the 204 μg sample of dry DNA contain a theoretical entropy of 1 bits/nucleotide × 60 nucleotides/strand × $4 \times 10^{15}$ strands = 28 PB (if there is no bias in the data), and 7 PB of randomness when calculating the ROV with 25% de-biasing efficiency.

Integrating the cost per synthesis run and the time for synthesis, the overall synthesis cost as well as the synthesis speed were calculated. For sequencing, scaled costs were calculated for the NovaSeq 6000 system with an S4 flow cell of $2 \times 100$ bp reads, which allows for 20 billion sequence reads in 36 h for a cost of 22,000 USD. We have evaluated sequencing output by the DNA trimming and selection scheme depicted in Supplementary Information 1. The overall sequencing cost as well as the sequencing speed were calculated.

**Reporting summary.** Further information on research design is available in the Nature Research Reporting Summary linked to this article.

## Data availability

The sequencing data underlying Fig. 3 is available on figshare repository: https://doi.org/10.6084/m9.figshare.12941786.v1. Any additional data will be made available upon reasonable request. Source data are provided with this paper.

## Code availability

The analysis code that supports the findings of this study is available upon request/is on figshare repository: https://doi.org/10.6084/m9.figshare.12941795.v1.

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

## Acknowledgements

The authors thank ICB/ETH Zurich and Microsoft for funding, and the Beat Christen Group at ETH for giving access to the iSeq 100 sequencer, as well as Fabian Axthelm and Markus Schmid from Microsynth AG, Switzerland for constant advice and fruitful discussions.

## Author contributions

R.N.G. and W.J.S. initiated and supervised the project. L.C.M. designed and performed the computational experiments, analyzed the data, performed sequencing analysis, and prepared all illustrations. J.K. sequenced DNA. R.N.G., P.L.A., and R.H. helped analyze the data. L.C.M. wrote the manuscript with assistance from R.H. and R.N.G. and input and approval from all authors.

## Competing interests

The authors declare no competing interests.
