## [Peer Review File · Nature Communications]

Reviewers' Comments:

Reviewer #1:

Remarks to the Author:

In this manuscript, Meiser et al. present a molecular form of Random Number Generation (RNG) using DNA synthesis and sequencing technologies. The major contribution of this work is an end-to-end experimental demonstration that these technologies are indeed capable of generating "random-enough" bit strings.

Overall, the novelty of the work is undermined by previous work that proposed and simulated the use of DNA synthesis and sequencing technologies for RNG, specifically see

Bogard et al. Random number generation for DNA-based security circuitry. BMC Bioinformatics 9, P13 (2008).

and

Gearheart et al. DNA-based random number generation in security circuitry. Biosystems. 100, 3. 2010.

It is also surprising that these previous works were not cited in this manuscript, even though a google search of "DNA random number generator" retrieves these papers in the top results. This being the case, this manuscript is still, to my knowledge, the first experimental demonstration of using DNA technologies for RNG.

I don't have any major issues with the experimental design and analysis methods from what I can read in the main text, however I am very surprised to not find a dedicated "Methods" or similar section of the manuscript? Surely a formal section that documents specific procedures and materials used is needed for reproducibility (and proper review) of the work.

There are also some other questions I had that I feel require attention if this manuscript goes through to publication:

1. The "error-free" sequence post-processing steps seems overly cumbersome. The authors divide the process into two trimming steps of which the logic seems arbitrary. Can they provide more explanation on why this trimming method was used? For example, why didn't the authors search each read for the forward and reverse adapter and extract the intervening sequence region. If the extracted region is not the correct length (eg 64 or 60 nt), discard it, and move to the next read.
2. Relatedly, why the authors synthesized a region of 64 Ns, but used a trimmed region of only 60 Ns for analysis is not discussed.
3. The authors should also discuss or at least reference the use of random nt sequences in other molecular barcoding apps, such as Universal Molecular Identifiers in NGS.
4. How the different block sizes for the NIST evaluation tests were chosen is not discussed. Also, did the authors show how different sizes change the result?
5. Have the authors considered if nucleotide-specific degradation (e.g. depurination) of the DNA over time could bias the strand sequences? For instance, the authors make throughput calculations based on the idea that many sequences can be synthesized in parallel and stored for long periods – the "randomness" of the strands may degrade over time.
6. Are sequencing errors (eg Illumina machine) error profiles truly random? Other references seem to suggest otherwise. For example, some error types are more/less common, and this would also be sequencing chemistry and machine dependent. For one reference see: Ma et al. "Analysis of error profiles in deep next-generation sequencing data" 2019.
7. I think it would be a great addition to this paper to include a table or plot comparing the methods used in this study to other RNG methods in terms of different factors, such as throughput and cost, latency, etc. I know the authors make some comparisons throughout the text, but it

would be nice to have these synthesized into an easily digestible figure to see how everything stacks up.

Reviewer #2:

Remarks to the Author:

Meiser et al. present a new approach to using synthetic DNA for the generation of true random numbers. Their method exploits the stochastic nature of the chemistry that is the basis to modern DNA synthesis technology. By incorporating mixed DNA bases in the synthesis process the resulting DNA contains a large pool of unique sequences sampled from a very high dimensional space. This approach is novel and has strong foundations in technological, chemical and algorithmic aspects. The authors demonstrate the suggested method with a series of experiments, followed by thorough analyses and computational steps.

The paper is well written and is of interest to diverse audience. The work is original and constitutes an important aspect of how synthetic DNA can be used, at least at the conceptual level.

However, there are several key points that should be addressed:

The manuscript lacks a proper methods section. Most of the methodology used in the project is described throughout the manuscript as part of the results or in figure captions. However, we believe that this makes following the experimental procedures and the analyses much harder. Also, to keep the manuscript easy to read, some descriptions lack essential technical information. We believe that merging all these into a single comprehensive methods section will greatly improve the readability of the paper while also allowing the authors to include all necessary details.

Some concrete examples:

The authors did not describe a complete protocol of the molecular biology procedures performed on the synthetic DNA to prepare it to NGS. Also, full details on the NGS run should be included. Was it a single-end or paired-end sequencing? How many bases were sequenced? What software tools (incl. versions and parameters) were used in the analysis?

The details described in the caption of Figure 4 don't belong in a figure caption but rather should be included in the missing methods section.

The calculation of the cost and rate should also move to a dedicated methods section and include more details.

The stated cost of 0.000014 USD/GB of randomness that appears in the abstract includes only the DNA synthesis cost. This cannot be stated as the total cost to produce randomness as all the other steps are essential steps for the generation of random numbers using the suggested method. Also, the cost and rate of randomness generation are not compared to current technologies.

This numerical detail should be removed from the abstract or replaced by a value that takes into account the cost of a complete end-to-end system for the generation of random numbers.

Moreover – it will be good to tie this to a specific use case example and estimate cost there.

Figure 4 is wasted real estate. The figure doesn't convey any information about the actual procedure of processing the DNA sequences.

This figure should be replaced with a more informative figure.

To assess the actual information leakage the authors can maybe implement a cryptographic protocol, such as secret sharing, say (Shamir 1979:

<https://dl.acm.org/doi/10.1145/359168.359176>) , and compare the distribution generated by using r produced by DNA to that generated by other random mechanisms. This can be done together with additional experiments, as below.

The very few experiments conducted to demonstrate the generation of random numbers using DNA synthesis included a single experiment for each experimental condition. It would be worthwhile to consider performing another experiment in which a number of technical repeats are

compared to assess the robustness of the system.

In Table 1, the P-value column should be explained. How should the numbers be interpreted? How does this compare to other sources of randomness.

Minor issues:

In page 2, the term "high quality random numbers" is too vague and should be explained.

In page 2, the last sentence in the main paragraph is missing a word "they are less prone to cryptanalysis attacks"

In Figure 2, the term $r_A, r_C, r_G, r_T, k_A, k_C, k_G, k_T, c_A, c_C, c_G, c_T$ are not defined. They do not appear anywhere else in the manuscript either.

In page 6, the paragraph describing the different biases in the synthetic DNA may be improved by referring to other studies that examined the technical properties of synthetic DNA molecules.

In page 6, the paragraph describing the different biases in the synthetic DNA should also refer to the two separate sources of bias – synthesis and sequencing. It might be worthwhile to try and distinguish synthesis based bias from sequencing based biases.

In page 6, the paragraph describing the different biases in the synthetic DNA should include analysis of biases in pairs of bases. This is specifically relevant in light of the "2 bits mapping scheme" that is analyzed later. This may be visualized using a 60x16 heatmap where rows are different base dimers, columns are positions and intensity represent the frequency.

In Figure 3, the lines in panels a-c are too thin.

In page 9, the authors refer to the random nature of sequencing errors. They state the errors occur in random positions. Are those errors also random in terms of the bases they affect? Are there errors that are more prone to occur?

In Figure 4 caption, the authors seem to be switching between the terms "primer" and "adapter". This should be clarified.

In Page 12, the bottom paragraph needs to be rephrased.

In page 12, the terms "computational efficiency" and "perfect bias removal" should be defined and the claims supported.

In Table 1, the caption sentence should be corrected. "NIST statistical tests performed on processed synthesized DNA oligonucleotides ..."

In Figure 6, panel c is the bit distribution after de-biasing. It is not clear how the bit stream length is still 60 bits after de-biasing. Is this due to some sort of splitting of the larger concatenated bit blocks? This should be clarified. Similarly, in the Sup material regarding the 2-bits-1-nucleotide mapping should contains 120 bits per raw bit stream but it shows only 60 bits.

There are multiple erroneous and unclear references. See for example R2, R27-29. Please revise the references. This is a major presentation flaw

Response to reviewers

Reviewer #1:

In this manuscript, Meiser et al. present a molecular form of Random Number Generation (RNG) using DNA synthesis and sequencing technologies. The major contribution of this work is an end-to-end experimental demonstration that these technologies are indeed capable of generating “random-enough” bit strings.

Overall, the novelty of the work is undermined by previous work that proposed and simulated the use of DNA synthesis and sequencing technologies for RNG, specifically see

Bogard et al. Random number generation for DNA-based security circuitry. BMC Bioinformatics 9, P13 (2008).

and

Gearheart et al. DNA-based random number generation in security circuitry. Biosystems. 100, 3. 2010.

It is also surprising that these previous works were not cited in this manuscript, even though a google search of “DNA random number generator” retrieves these papers in the top results. This being the case, this manuscript is still, to my knowledge, the first experimental demonstration of using DNA technologies for RNG.

I don’t have any major issues with the experimental design and analysis methods from what I can read in the main text, however I am very surprised to not find a dedicated “Methods” or similar section of the manuscript? Surely a formal section that documents specific procedures and materials used is needed for reproducibility (and proper review) of the work.

We would like to thank the referee for his/her constructive criticism. We have implemented a comparison to previous work with respect to the individual points and would like to address them separately below.

We have now cited the two works by Bogard et al. and Gearheart et al., as of course these show relevant initial findings in the field of DNA random number generation. The simulations presented in these works show the idea of DNA as a random number generation nicely, but unfortunately do not show the encountered hurdles (such as bias in synthesis data), which appear when realizing DNA random number generation experimentally, as we have shown.

We have added the reference in the manuscript as follows:

“Previous work has presented the idea of simulation of the DNA random number generation circuitry by theoretically proposing a scheme for a possible automated workflow for DNA random number generation. However, the physical realization of the theory and the limitations experimentally were not investigated.” (Bogard et al. Random number generation for DNA-based security circuitry. BMC Bioinformatics 9, P13 (2008). and Gearheart et al. DNA-based random number generation in security circuitry. Biosystems. 100, 3. 2010.)

We have also added a relevant METHODS section following the conclusion of this work. As the reviewer has suggested, this section now specifies all procedures and materials used.

There are also some other questions I had that I feel require attention if this manuscript goes through to publication:

1. The “error-free” sequence post-processing steps seems overly cumbersome. The authors divide the process into two trimming steps of which the logic seems arbitrary. Can they provide more explanation on why this trimming method was used? For example, why didn’t the authors search each read for the forward and reverse adapter and extract the intervening sequence region. If the extracted region is not the correct length (eg 64 or 60 nt), discard it, and move to the next read.

We have added a more holistic explanation detailing the purpose of the procedure shown in the new METHODS section of our work:

“**Selection of Error-Free Sequences.** To only allow for sequences that have been synthesized and sequenced correctly, sequences containing 16 nucleotides of the adapter (AGA TCG GAA GAG CAC A) were searched for and used for further data treatment. All other sequences were discarded. The remaining sequences were then shortened to 69 nucleotides. As deletions may occur during synthesis and sequencing steps, the designed random region of 64 nucleotides may be shorter than designed. Thus, the remaining 69-nucleotide long sequences were searched for the first 9 nucleotides of the adapter (AGA TCG GAA). This time, all sequences still containing these nucleotides were discarded from the pool. All other sequences were shortened to 60 nucleotides. Thus, it is guaranteed, that the random region does not contain any adapter nucleotides due to deletion errors. An illustration of this procedure can be found in Supplementary Information 1.”

We have additionally expanded Supplementary Information 1 with further explanations with a new illustration to visualize the procedure:

- nucleotide of random sequence
- nucleotide of adapter
- ✓ keep sequence
- ✗ discard sequence

- (1) cut all sequences containing 16 nucleotides of adapter at 69 nucleotides
- (2) search for sequences containing 9 adapter nucleotides and delete those sequences
- (3) shorten all other sequences to 60 nucleotides

Supplementary Figure 1b: procedure of selecting error-free sequences. Previously to this, the pool of DNA sequences has been searched for sequences containing 16 nucleotides of the adapter (AGA TCG GAA GAG CAC A). Sequences not containing these 16 nucleotides of adapter were discarded from the pool.

The trimming method is set to remove any bias from adapters faultily entering the random region due to deletion errors. This would not be the case, if we only scan the DNA strands for each adapter once, as suggested by the reviewer, as shorter parts of the adapter may still be in the expected random region due to deletion errors within sequences.

2. Relatedly, why the authors synthesized a region of 64 Ns, but used a trimmed region of only 60 Ns for analysis is not discussed.

As already mentioned in the response to the reviewer's previous remark, we have now added an extensive explanation to the METHODS section detailing why the trimming as presented is a very suitable method for error removal. The additional illustration (see response to previous remark) visualizes the trimming process further, and is now shown in Supplementary Information 1. As deletion errors can trim the 64-nucleotide random section of randomness, we have shortened the synthesized 64-nucleotide oligos to 60-nucleotide oligos all containing random bases.

3. The authors should also discuss or at least reference the use of random nt sequences in other molecular barcoding apps, such as Universal Molecular Identifiers in NGS.

We thank the reviewer for this insight to improve our work. We have included information about the use of random DNA in terms of barcoding in the section "Design of DNA", where we also mention another application of random synthesis relating to taxonomic classification. The new sentence reads:

"Other applications for random nucleotide syntheses are found in barcoding, where, by means of unique molecular identifiers (UMI), PCR amplification bias can be eliminated." (Kivioja, T. *et al.* Counting absolute numbers of molecules using unique molecular identifiers. *Nat. Methods* **9**, 72–74 (2012).)

4. How the different block sizes for the NIST evaluation tests were chosen is not discussed. Also, did the authors show how different sizes change the result?

We thank the reviewer for this comment. We have added a METHODS section dedicated to *NIST Evaluation Parameters*, in which we give the parameter choice and note any parameter adjustments that have been undertaken:

"NIST Evaluation Parameters. NIST evaluation tests were chosen such that for each test, 56 bit streams containing 1096 bits were tested (with the exception of the rank test, where 56 bit streams containing 100,000 bits each were tested). This variant of statistical analyses was chosen and explained by Rojas *et. al.*¹⁴ The tests were applied with the standard parameters as given by the NIST statistical test suite,¹² with the following parameters differing from set values: 1) frequency test, no parameter adjustments; 2) block frequency test, no parameter adjustments (block length, $M = 128$); 3) cumulative sums test, no parameter adjustments; 4) runs test, no parameter adjustments; 5) longest runs of ones test, no parameter adjustments; 6) rank test, no parameter adjustments; 7) discrete Fourier transform test, no parameter adjustments; 8) approximate entropy test parameter adjustment: block length, $m = 5$; 9) serial test no parameter adjustments (block length, $m = 16$). For each statistical test, the NIST software computes a P-value, which gives the probability that the sequence tested is more random than the sequence a perfect random number generator would have produced. Thereby, a P-value of 1 indicates perfect randomness, whereas a P-value of 0 indicated complete non-randomness. More specifically, for a P-value ≥ 0.001 : sequence can be considered random with a confidence of 99.9%, and for a P-value < 0.001 sequence can be considered non-random with a confidence of 99.9%." (Rukhin, A., Soto, J., Nechvatal, J. & Smid, M. A Statistical Test Suite for Random and Pseudorandom Number Generators for Cryptographic Applications. *NIST* (2010). and Gaviria Rojas, W. A. *et al.* Solution-processed carbon nanotube true random number generator. *Nano Lett.* **17**, 4976–4981 (2017).)

We show that with the choice of these parameters, DNA synthesis can be applied for random number generation. We have altered the parameters and obtained the same result. We have further performed a series of randomness evaluations using NIST statistical test suite, in which the robustness of the system was assessed with different technical parameters. We investigated different numbers of bit streams as well as

different lengths of bit streams and observed the following: With 56 bit streams, each of length 10,000 bits, all tests pass. The shorter the bit stream length tested, the worse the Rank test performs and the longer the bit stream length tested, the worse the Runs test performs (with the performance of all other tests being similar in all cases). When increasing the number of bit streams investigated, the Runs test performance decreases. At high bit stream numbers and bit stream lengths, the Discrete Fourier Transform test starts showing less robust results. All other tests show very robust evaluation performance at all values of bit stream length and number of bit streams tested.

We have added this information to our manuscript as follows:

“Robustness evaluation of the NIST statistical test suite with respect to our data is shown in Supplementary Information 3.”

And in Supplementary Information 3:

“We have performed a series of randomness evaluations using NIST statistical test suite, in which the robustness of the system was assessed with different technical parameters. We investigated different numbers of bit streams as well as different lengths of bit streams and observed the following: With 56 bit streams, each of length 10,000 bits, all tests pass. The shorter the bit stream length tested, the worse the Rank test performs and the longer the bit stream length tested, the worse the Runs test performs (with the performance of all other tests being similar in all cases). When increasing the number of bit streams investigated, the Runs test performance decreases. At high bit stream numbers and bit stream lengths, the Discrete Fourier Transform test starts showing negative results. All other tests show very robust evaluation performance at all values of bit stream length and number of bit streams tested.

Table: Robustness of NIST statistical tests performed on Von Neumann processed bit streams, mapped to bits using the scheme $A \rightarrow 0, C \rightarrow 0, T \rightarrow 1, G \rightarrow 1$.”

Number of bit streams	56	56	56	100	100	100
Length of bit streams	50,000	10,000	1,096	50,000	10,000	1,096
Tests failing	Runs	All tests pass	Rank	Runs	Runs close to failing	Rank, Discrete Fourier transform close to failing

5. Have the authors considered if nucleotide-specific degradation (e.g. depurination) of the DNA over time could bias the strand sequences? For instance, the authors make throughput calculations based on the idea that many sequences can be synthesized in parallel and stored for long periods – the “randomness” of the strands may degrade over time.

It has indeed been shown that DNA is prone to chemical decay, especially hydrolytic damage, which results in depurination and eventually strand break. Using current Illumina sequencing technologies usually involves DNA amplification steps using PCR for sample preparation. Any strand that has experienced strand breakage during storage is no longer read when the pool of DNA is being sequenced. This is due to the fact that the broken DNA strands do not contain the correct priming sites for PCR amplification, which leads to those sequences not containing the correct sequencing adapters for the sequences to be read. They are thus diluted out of the pool of DNA (Heckel, R. *et.al.* A Characterization of the DNA Data Storage Channel. *Sci.*

Rep. **9**, 1–12 (2019).). If the pool of randomness that has been generated initially is stored adequately by encapsulating it in an inorganic matrix (as illustrated by Grass, R. N. *et.al.* Robust chemical preservation of digital information on DNA in silica with error-correcting codes. *Angew. Chem. Int. Ed.* **54**, 2552–2555 (2015).), DNA degradation is minimized. However, if sequences shall still break during storage, the loss of random sequences solely reduces the volume of randomness remaining after storage. To prove this point, we have simulated a 50%-loss of random sequences from the overall pool and show that this pool, too, passes the NIST evaluation of randomness with the following parameters (after being processed according to our defined processing scheme):

Table: NIST Statistical Test Suite Randomness Evaluation. NIST statistical tests performed on processed synthesized DNA oligonucleotides, mapped to bits using the scheme A → 0, C → 0, T → 1, G → 1. For each test, 56 bit streams containing 1096 bits were tested.

NIST statistical test	P-value	Proportion	Result
Frequency	0.851383	56/56	Pass
Block frequency	0.383827	56/56	Pass
Cumulative sums	0.779188	56/56	Pass
Runs	0.153763	53/56	Pass
Longest run of ones	0.066882	56/56	Pass
Rank	0.066882	56/56	Pass
Discrete Fourier transform	0.699313	55/56	Pass
Approximate entropy	0.213309	56/56	Pass
Serial	0.779188	56/56	Pass

6. Are sequencing errors (eg Illumina machine) error profiles truly random? Other references seem to suggest otherwise. For example, some error types are more/less common, and this would also be sequencing chemistry and machine dependent. For one reference see: Ma et al. “Analysis of error profiles in deep next-generation sequencing data” 2019.

We thank the reviewer for this valuable thought. It is indeed the case, that sequencing errors may occur. These are mainly of non-random nature, when phasing effects are not accounted for (Pfeiffer, F. et al. Systematic evaluation of error rates and causes in short samples in next-generation sequencing. *Sci. Rep.* **8**, 1–14 (2018)). We do account for phasing effects, as we select sequences with the right sequence length and shorten all these “correct” sequences to 60 nucleotides. However, we show that even though non-random sequencing errors may occur (as suggested in the reference you provide by Ma et al.), our system of debiasing (using the Von Neumann algorithm as shown) is robust to account for such non-random errors. We have elaborated our discussion on biases and sequencing errors. It now reads as follows:

“Pfeiffer et. al. have shown that after processing the DNA pool by means of sequencing-by-synthesis using Illumina, removing (faulty) shortened sequences, errors from sequencing DNA occur at seemingly random positions. Other studies have shown that errors may be related to the sequencing context, where the highest rate of errors was

found for nucleotide T, which in our case would result to be erroneous once in every 1,250 nucleotides, and the error rates for C, G, and A being much lower. Overall, expected error rates were found to increase towards the end of the DNA strand. To minimize the influence of (especially) deletion errors on randomness, we shortened all sequences to 60 nucleotides and simultaneously selected only the sequences containing the correct length of random nucleotides (Supplementary Information 1).” (Pfeiffer, F. *et al.* Systematic evaluation of error rates and causes in short samples in next-generation sequencing. *Sci. Rep.* **8**, 1–14 (2018). and Schirmer, M., D’Amore, R., Ijaz, U. Z., Hall, N. & Quince, C. Illumina error profiles: Resolving fine-scale variation in metagenomic sequencing data. *BMC Bioinformatics* **17**, 1–15 (2016).)

7. I think it would be a great addition to this paper to include a table or plot comparing the methods used in this study to other RNG methods in terms of different factors, such as throughput and cost, latency, etc. I know the authors make some comparisons throughout the text, but it would be nice to have these synthesized into an easily digestible figure to see how everything stacks up.

We thank the reviewer for this helpful suggestion to improve the quality of our work. We have compiled several commercial and academic methods for randomness generation and illustrate these in a table we call “Table 2”. Together with a short discussion, we have integrated this into our manuscript as follows:

“In comparison to other random number generation methods (Table 2), DNA synthesis shows a higher randomness production rate than many commercial options such as the online distributors Random.org or HotBits, for example.^{45,46} However, randomness generation can also be orders of magnitudes faster, especially when shifting from true random number generators to pseudo random number generators. When using random numbers transmitted through the internet it is important to consider that these come with limitations such as potential interception of data, which adds a certain degree of uncertainty to the security.”

Random generator	number	Randomness production rate [MB/s]	Method
Meiser et. al.		0.3	DNA synthesis
Gaviria Rojas et. al.		Not available	Solution-processed carbon nanotubes
Lee et. al.		0.025	Crystallization robot analyzing chemical processes
Reidler et. al.		1,560	Chaotic semiconductor laser
HotBits		0.0001	Timing successive pairs of radioactive decays
Random.org		0.0015	Entropy from atmospheric noise
Lavarnd		0.02	Patterns photographed off floating material in lava lamps
Intel digital random number generator		800	Processor resident entropy source to seed hardware-implemented entropy from atmospheric noise
Mersenne Twister		15,000*	*Pseudo random number generator: algorithm using polynomial algebra

(Noll, L. C., Mende, R. G. & Sisodiya, S. Method for seeding a pseudo-random number generator with a cryptographic hash of a digitization of a chaotic system. *United States Patent* (1998). and Matsumoto, M. & Nishimura, T. Mersenne Twister: A 623-Dimensionally Equidistributed Uniform Pseudo-Random Number Generator. *ACM Trans. Model. Comput. Simul.* **8**, 3–30 (1998). and Jun, B. & Kocher, P. The Intel Random Number Generator. *Cryptogr. Res.* 1–8 (1999). and Gaviria Rojas, W. A. *et al.* Solution-processed carbon nanotube true random number generator. *Nano Lett.* **17**, 4976–4981 (2017). and Lee, E. C., Parrilla-Gutiérrez, J. M., Henson, A., Brechin, E. K. & Cronin, L. A Crystallization Robot for Generating True Random Numbers Based on Stochastic Chemical Processes. *Matter* **2**, 1–9 (2020). and Haar, M. RANDOM.ORG: True Random Number Service. <https://www.random.org/> (1998). and Walker, J. HotBits: Genuine random numbers, generated by radioactive decay.

<https://www.fourmilab.ch/hotbits/> (1996). and Reidler, I., Aviad, Y., Rosenbluh, M. & Kanter, I. Ultrahigh-speed random number generation based on a chaotic semiconductor laser. *Phys. Rev. Lett.* **103**, 1–4 (2009.)

Reviewer #2 (Remarks to the Author):

Meiser et al. present a new approach to using synthetic DNA for the generation of true random numbers. Their method exploits the stochastic nature of the chemistry that is the basis to modern DNA synthesis technology. By incorporating mixed DNA bases in the synthesis process the resulting DNA contains a large pool of unique sequences sampled from a very high dimensional space. This approach is novel and has strong foundations in technological, chemical and algorithmic aspects. The authors demonstrate the suggested method with a series of experiments, followed by thorough analyses and computational steps. The paper is well written and is of interest to diverse audience. The work is original and constitutes an important aspect of how synthetic DNA can be used, at least at the conceptual level. However, there are several key points that should be addressed:

The manuscript lacks a proper methods section. Most of the methodology used in the project is described throughout the manuscript as part of the results or in figure captions. However, we believe that this makes following the experimental procedures and the analyses much harder. Also, to keep the manuscript easy to read, some descriptions lack essential technical information.

We believe that merging all these into a single comprehensive methods section will greatly improve the readability of the paper while also allowing the authors to include all necessary details. Some concrete examples:

The authors did not describe a complete protocol of the molecular biology procedures performed on the synthetic DNA to prepare it to NGS. Also, full details on the NGS run should be included. Was it a single-end or paired-end sequencing? How many bases were sequenced? What software tools (incl. versions and parameters) were used in the analysis?

The details described in the caption of Figure 4 don't belong in a figure caption but rather should be included in the missing methods section.

The calculation of the cost and rate should also move to a dedicated methods section and include more details.

We would like to thank the referee for his/her positive feedback and all the constructive input. We have implemented the suggestions and feel like the quality of our manuscript has greatly improved since. We would like to address each point and our respective implementations separately below.

Methods section: We have added a METHODS sections to give a complete protocol of all the chemical, biological and computational procedures. In this new section, we have added a paragraph dedicated to calculations of cost and rate for randomness generation, in which we give more detail on the individual calculations and assumptions performed. Sample preparation and next-generation sequencing parameters are also shown in respective paragraphs, as suggested by the reviewer. We give all information about software tools (including versions) used for this analysis. The section of the methods titled “Next-Generation Sequencing” now reads as follows:

“Next-Generation Sequencing. Each sample was diluted to 1 nM and then further processed for sequencing using the iSeq100 Sequencing System Guide. For quality control, 2% (vol/vol) PhiX were added to the sequencing run. (PhiX is a reliable, adapter-ligated, ready-to-use genomic DNA sequencing control library provided by Illumina). Sequencing was performed on Illumina’s iSeq 100 with a paired-end read length of 2 ×

150 bp. Analysis of sequencing output was performed using Matlab version R2018b and python version 3.8.”

Figure 4: We have taken figure 4 and its caption out of the main text and moved it to Supplementary Information 1. The information from the figure caption has also been moved and detailed in the new METHODS section. The part of the methods section illustrating the procedure, which was originally found in figure 4 now reads as follows:

“Selection of Error-Free Sequences. To only allow for sequences that have been synthesized and sequenced correctly, sequences containing 16 nucleotides of the adapter (AGA TCG GAA GAG CAC A) were searched for and used for further data treatment. All other sequences were discarded. The remaining sequences were then shortened to 69 nucleotides. As deletions may occur during synthesis and sequencing steps, the designed random region of 64 nucleotides may be shorter than designed. Thus, the remaining 69-nucleotide long sequences were searched for the first 9 nucleotides of the adapter (AGA TCG GAA). This time, all sequences still containing these nucleotides were discarded from the pool. All other sequences were shortened to 60 nucleotides. Thus, it is guaranteed, that the random region does not contain any adapter nucleotides due to deletion errors. An illustration of this procedure can be found in Supplementary Information 1.”

Cost and rate calculation: As suggested by the reviewer, we have moved sections of the cost and rate calculations from the RESULTS AND DISCUSSION section to the METHODS section. We have added formulas and details for the calculations performed. The respective section now reads as follows:

“Calculations of Cost and Rate for Randomness Generation. Cost and rate of randomness generation were calculated as a basis of the number of random nucleotides synthesized. This was done by calculating the number of strands synthesized (NS) from the amount of DNA synthesized (M): $NS = M \times Avogadro\ constant$

For the number of random bits synthesized (NRB), 60 random nucleotides per strand were assumed. The randomness output volume (ROV) was calculated by assuming a 25% de-biasing efficiency: $ROV = 0.25 \times NRB$

The possible entropy of a random 60-mer is $4^{60} = 10^{36}$, and thus is significantly larger than the number of sequences in the pool. This implies that it may be expected that every strand synthesized is unique. As a result, the 204 μg sample of dry DNA contain a theoretical entropy of 1 bits/nucleotide \times 60 nucleotides/strand \times 4×10^{15} strands = 28 PB (if there is no bias in the data), and 7 PB of randomness when calculating the ROV with 25% de-biasing efficiency.

Integrating the cost per synthesis run and the time for synthesis, the overall synthesis cost as well as the synthesis speed were calculated. For sequencing, scaled costs were calculated for the NovaSeq 6000 system with an S4 flow cell of 2×100 bp reads, which allows for 20 billion sequence reads in 36 h for a cost of 22,000 USD. We have evaluated sequencing output by the DNA trimming and selection scheme depicted in Supplementary Information 1. The overall sequencing cost as well as the sequencing speed were calculated.”

The stated cost of 0.000014 USD/GB of randomness that appears in the abstract includes only the DNA synthesis cost. This cannot be stated as the total cost to produce randomness as all the other steps are essential steps for the generation of random numbers using the suggested method. Also, the cost and rate of randomness generation are not compared to current technologies.

This numerical detail should be removed from the abstract or replaced by a value that takes into account the cost of a complete end-to-end system for the generation of random numbers. Moreover – it will be good to tie this to a specific use case example and estimate cost there.

We have replaced the cost of DNA synthesis in the abstract with the overall random number generation cost. The respective sentence in the abstract has now been changed to the following:

“We compare three commercial random DNA syntheses giving a measure for robustness and synthesis distribution of nucleotides and show that using DNA for random number generation, we can obtain 7 million GB of randomness from one synthesis run, which can be read out using state-of-the-art sequencing technologies at rates of ca. 300 kB/s.”

Additionally, to address the reviewer’s point of comparing randomness generation to current technologies, we have added a table (called “Table 2” in our results section), summarizing the rates of random number generation by different methods. This table features the random number generators cited throughout our text, as well as some online and commercial state-of-the-art random number generators for throughput comparison:

Random generator	number	Randomness production rate [MB/s]	Method
Meiser et. al.		0.3	DNA synthesis
Gaviria Rojas et. al.		Not available	Solution-processed carbon nanotubes
Lee et. al.		0.025	Crystallization robot analyzing chemical processes
Reidler et. al.		1,560	Chaotic semiconductor laser
HotBits		0.0001	Timing successive pairs of radioactive decays
Random.org		0.0015	Entropy from atmospheric noise
Lavarnd		0.02	Patterns photographed off floating material in lava lamps
Intel digital random number generator		800	Processor resident entropy source to seed hardware-implemented entropy from atmospheric noise
Mersenne Twister		15,000*	*Pseudo random number generator: algorithm using polynomial algebra

(Noll, L. C., Mende, R. G. & Sisodiya, S. Method for seeding a pseudo-random number generator with a cryptographic hash of a digitization of a chaotic system. *United States Patent* (1998). and Matsumoto, M. & Nishimura, T. Mersenne Twister: A 623-Dimensionally Equidistributed Uniform Pseudo-Random Number Generator. *ACM Trans. Model. Comput. Simul.* **8**, 3–30 (1998). and Jun, B. & Kocher, P. The Intel Random Number Generator. *Cryptogr. Res.* 1–8 (1999). and Gaviria Rojas, W. A. et al. Solution-processed carbon nanotube true random number generator. *Nano Lett.* **17**, 4976–4981 (2017). and Lee, E. C., Parrilla-Gutiérrez, J. M., Henson, A., Brechin, E. K. & Cronin, L. A Crystallization Robot for Generating True Random Numbers Based on Stochastic Chemical Processes. *Matter* **2**, 1–9 (2020). and Haar, M. RANDOM.ORG: True Random Number Service. <https://www.random.org/> (1998). and Walker, J. HotBits: Genuine random numbers, generated by radioactive decay. <https://www.fourmilab.ch/hotbits/> (1996). and Reidler, I., Aviad, Y., Rosenbluh, M. & Kanter, I. Ultrahigh-speed random number generation based on a chaotic semiconductor laser. *Phys. Rev. Lett.* **103**, 1–4 (2009).)

Figure 4 is wasted real estate. The figure doesn’t convey any information about the actual procedure of processing the DNA sequences. This figure should be replaced with a more informative figure.

We have removed this Figure 4 from the main text, and added it to the Supplementary Information document. We have not replaced the figure with another figure, but have instead added “Table 2” for technology comparison, as explained previously. The new table provides context to where the DNA random

number generation technology stands commercially and academically and adds great value to our work. We thank the reviewer for this suggestion.

To assess the actual information leakage the authors can maybe implement a cryptographic protocol, such as secret sharing, say (Shamir 1979: <https://dl.acm.org/doi/10.1145/359168.359176>), and compare the distribution generated by using r produced by DNA to that generated by other random mechanisms. This can be done together with additional experiments, as below. The very few experiments conducted to demonstrate the generation of random numbers using DNA synthesis included a single experiment for each experimental condition. It would be worthwhile to consider performing another experiment in which a number of technical repeats are compared to assess the robustness of the system.

We have expanded the set of experiments evaluating our random number generation technology and performed a series of assessment tests using NIST statistical test suite, in which the robustness of the system was assessed with different technical parameters, as suggested by the reviewer. We investigated different numbers of bit streams as well as different lengths of bit streams and observed the following: With 56 bit streams, each of length 10,000 bits, all tests pass. The shorter the bit stream length tested, the worse the Rank test performs and the longer the bit stream length tested, the worse the Runs test performs (with the performance of all other tests being similar in all cases). When increasing the number of bit streams investigated, the Runs test performance decreases. At high bit stream numbers and bit stream lengths, the Discrete Fourier Transform test starts showing negative results. All other tests show very robust evaluation performance at all values of bit stream length and number of bit streams tested.

We have added this information to our manuscript as follows:

“Robustness evaluation of the NIST statistical test suite with respect to our data is shown in Supplementary Information 3.”

And in Supplementary Information 3:

“We have performed a series of randomness evaluations using NIST statistical test suite, in which the robustness of the system was assessed with different technical parameters. We investigated different numbers of bit streams as well as different lengths of bit streams and observed the following: With 56 bit streams, each of length 10,000 bits, all tests pass. The shorter the bit stream length tested, the worse the Rank test performs and the longer the bit stream length tested, the worse the Runs test performs (with the performance of all other tests being similar in all cases). When increasing the number of bit streams investigated, the Runs test performance decreases. At high bit stream numbers and bit stream lengths, the Discrete Fourier Transform test starts showing negative results. All other tests show very robust evaluation performance at all values of bit stream length and number of bit streams tested.

Table: Robustness of NIST statistical tests performed on Von Neumann processed bit streams, mapped to bits using the scheme $A \rightarrow 0, C \rightarrow 0, T \rightarrow 1, G \rightarrow 1$.”

Number of bit streams	56	56	56	100	100	100
Length of bit streams	50,000	10,000	1,096	50,000	10,000	1,096
Tests failing	Runs	All test pass	Rank	Runs	Runs close to failing	Rank, Discrete Fourier transform close to failing

The quality of a cryptographic protocol is partly determined by the quality of the randomness it uses. It is standard in the literature on random number generation to evaluate those two elements of a cryptographic systems separately, and to not mix them by studying the same cryptographic protocols for different sources of randomness. Comparing the output of a cryptographic protocol is not a standard way to study the quality of randomness, and we therefore decided against including such a comparison. Instead, we expanded our evaluation using the NIST statistical test suite as shown above (following the vast literature on random number generation), and rather extended testing of the robustness of our system using various parameters for the variety of tests for evaluating the quality of the randomness.

In Table 1, the P-value column should be explained. How should the numbers be interpreted? How does this compare to other sources of randomness.

Thank you for addressing the importance of the P-value. We have added the decision level based on the P-value to the text directly with more detail in the new METHODS section. NIST statistical test suite uses the following criteria: $P\text{-value} \geq 0.001$: sequence is random with 99.9% confidence, $P\text{-value} < 0.001$ sequence is non-random with 99.9% confidence. The P-value is independent of the source of randomness, and solely evaluates randomness based on the string of bits provided to the software.

Minor issues:

In page 2, the term “high quality random numbers” is too vague and should be explained.

We have addressed this point by changing the respective sentence as follows:

“Shifting from algorithm to interactions, the modern world required network security services, and thus introduced encryption and decryption schemes for exchanging information securely, requiring high quality random numbers (generated faster while being less prone to attacks)”

In page 2, the last sentence in the main paragraph is missing a word “they are less prone to cryptanalysis attacks”

We thank the reviewer for having drawn our attention to this missing word. We have now added the missing word “to” to the sentence:

“Such hardware random number generators create bit streams depending on highly unpredictable physical processes, making them useful for secure data transmission as they are less prone to cryptanalytic attacks.”

In Figure 2, the term $r_A, r_C, r_G, r_T, k_A, k_C, k_G, k_T, c_A, c_C, c_G, c_T$ are not defined. They do not appear anywhere else in the manuscript either.

We thank the reviewer for this comment. We have added the following information to the figure caption:

“The rate of the individual nucleotides coupling efficiencies, r_i , can be approximated by multiplication of the respective rate constant, k_i and the nucleotide concentration, c_i .»

In page 6, the paragraph describing the different biases in the synthetic DNA may be improved by referring to other studies that examined the technical properties of synthetic DNA molecules.

We have added a more comprehensive discussion to improve this paragraph, as suggested by the reviewer’s remark. Studies have investigated the effects of errors such as those induced by depurination on resulting oligo lengths We now mention these studies in a short discussion with a sentence as follows:

During synthesis of DNA, growing strands may be terminated before having reached the desired length and thus induce a bias to the pool. We account for this synthesis bias by

following strict sequence selection criteria.” (LeProust, E. M. *et al.* Synthesis of high-quality libraries of long (150mer) oligonucleotides by a novel depurination controlled process. *Nucleic Acids Res.* **38**, 2522–2540 (2010).).

In page 6, the paragraph describing the different biases in the synthetic DNA should also refer to the two separate sources of bias – synthesis and sequencing. It might be worthwhile to try and distinguish synthesis based bias from sequencing based biases.

We have added a discussion of sequencing and synthesis bias, in the RESULTS AND DISCUSSION, where we also mention the possibilities of coverage bias (due to PCR stochasticity or oligo distribution on synthesis chips) with a short analysis. For our system it is close to impossible to decouple synthesis from sequencing bias. We give a short discussion on this as well as the potential sources of errors from sequencing that have been shown in previous works:

“There are two main potential sources of bias that can have an effect on the results as shown in Figure 3: coverage bias and error bias. The former bias has been investigated by Chen *et al.* and is predominantly expressed by bias that can be related to the spatial location on the synthesis chip and PCR stochasticity. The latter bias is the result of insertion, deletion or substitution of erroneous nucleotides during synthesis, PCR and sequencing steps. For our work, coverage bias only influences the nucleotide distribution if there is a significant discrepancy between coverage of each random sequence. We have analyzed this by counting the number of occurrences of each sequence and found that a single sequence is not present in the pool more than five times with a mean presence of each sequence of 1.03 times. This implies that the bias from sequence coverage cannot be the reason for the observed *nucleotide nonequivalence* and *position nonequivalence* behavior. As for the error bias, it is difficult to distinguish between synthesis and sequencing errors as the two processes cannot be completely decoupled, as access to the molecular morphology of DNA is only possible through sequencing DNA. However, studies have suggested that if data is handled accordingly, sequencing errors occur at random positions. During synthesis of DNA, growing strands may be terminated before having reached the desired length and thus induce a bias to the pool. We account for this synthesis bias by following strict sequence selection criteria. To our best knowledge, no studies of sequencing error trends have shown effects of nature and magnitude as we see in Figures 3a-3c. We therefore must conclude that the trend seen is predominantly originating from synthesis.” (Chen, Y *et al.* Quantifying molecular bias in DNA data storage. *Nat Commun* **11**, 3264 (2020) and Pfeiffer, F. *et al.* Systematic evaluation of error rates and causes in short samples in next-generation sequencing. *Sci. Rep.* **8**, 1–14 (2018). and LeProust, E. M. *et al.* Synthesis of high-quality libraries of long (150mer) oligonucleotides by a novel depurination controlled process. *Nucleic Acids Res.* **38**, 2522–2540 (2010). and Heckel, R., Mikutis, G. & Grass, R. N. A Characterization of the DNA Data Storage Channel. *Sci. Rep.* **9**, 1–12 (2019).)

and

“Errors that may have occurred include deletion, insertion and substitution errors and can result in the DNA strand being short of a base, too long by a base or containing a faulty base, respectively. The selection procedure is illustrated in Supplementary Information 1. Pfeiffer *et al.* have shown that after processing the DNA pool by means of sequencing-by-synthesis using Illumina, removing (faulty) shortened sequences, errors from sequencing DNA occur at seemingly random positions. Other studies have shown that errors may be related to the sequencing context, where the highest rate of errors was found for nucleotide T, which in our case would result to be erroneous once in every 1,250 nucleotides, and the error rates for C, G, and A being much lower. Overall, expected error rates were found to increase towards the end of the DNA strand. To minimize the influence of (especially) deletion errors on randomness, we shortened all sequences to 60 nucleotides and simultaneously selected only the sequences containing the correct length of random nucleotides (Supplementary Information 1).” (Pfeiffer, F. *et*

al. Systematic evaluation of error rates and causes in short samples in next-generation sequencing. *Sci. Rep.* **8**, 1–14 (2018). and Meiser, L. C. *et al.* Reading and writing digital data in DNA. *Nat. Protoc.* **15**, 86–101 (2020). and Schirmer, M., D’Amore, R., Ijaz, U. Z., Hall, N. & Quince, C. Illumina error profiles: Resolving fine-scale variation in metagenomic sequencing data. *BMC Bioinformatics* **17**, 1–15 (2016.)

In page 6, the paragraph describing the different biases in the synthetic DNA should include analysis of biases in pairs of bases. This is specifically relevant in light of the “2 bits mapping scheme” that is analyzed later. This may be visualized using a 60x16 heatmap where rows are different base dimers, columns are positions and intensity represent the frequency.

We thank the reviewer for this very insightful suggestion to improve the research presented and to scientifically expand the investigation. We have thus performed an analysis of nucleotide pair distribution across the DNA sequence and have created a heat map showing the results for nucleotide pairs, normalized to the relative amounts of each nucleotide present in the strands. We have adjusted the figure caption respectively, and appended the METHODS with the procedure we followed during normalizing.

The heat map now shows the prevalence of two nucleotides binding, given the nucleotide to which to bind to:

“By normalization Microsynth synthesis 1 (Fig 3a), we obtain a heat map illustrating the prevalence of two nucleotides binding (Fig 3d), and can observe a third bias: *Nucleotide binding prevalence*. We see that the preference for one base binding to the existing nucleotide is partially dependent on the nature of the existing nucleotide, thus, guanine is least prevalent to bind to an adenine (normalized proportion < 0), if it has the possibility to bind to an adenine, thymine, cytosine or guanine, and guanine is most prevalent to bind to guanine (normalized proportion > 0), if it is free to bind to adenine, thymine, cytosine or guanine. Note that the nucleotide pair distribution cannot only be explained by the position-dependent bias as seen in Figure 3a: For example, by just going by bias CA should be less frequent than GG at the beginning of the sequence, yet they are equally likely.”

“Figure 3: Commercial DNA Syntheses. Commercial DNA synthesis of random nucleotides from (a) Microsynth, (b) Microsynth, (c) Eurofins Genomics and the mechanism of transversion (d) heat map showing normalized nucleotide pair distribution along the strand length of Microsynth 1 synthesis. (e). For the three syntheses, illustrated are the 60 nucleotides of the random section of DNA strands synthesized, respectively,

based on the design featured in Figure 1. The direction of synthesis is indicated by arrows. Analyzed sample size ca. 700,000 sequences each. The mechanism of G-T transversion can be a source for the effect of percentage variation of G and T along the strand of DNA (position nonequivalence)”

We believe that this addition to our investigation has significantly improved the quality of our manuscript and our understanding of the process of synthesizing DNA.

In Figure 3, the lines in panels a-c are too thin.

We have changed the thickness of the lines. The figure looks as shown in the previous remark's response.

In page 9, the authors refer to the random nature of sequencing errors. They state the errors occur in random positions. Are those errors also random in terms of the bases they affect? Are there errors that are more prone to occur?

Thank you for these valuable questions to investigate. We have added a few sentences discussing the nature of sequencing errors. Schirmer et al. (now cited) have, in their study, found slightly higher error rates for T (0.0008 errors per base) than for G (0.0005 errors per base), and even slightly lower error rates for A and C (0.0004 errors per base). However, positional error investigations remained in part inconclusive. (Schirmer et al. Illumina error profiles: resolving fine-scale variation in metagenomic sequencing data. *BMC Bioinformatics* **17**, 125 (2016).) However, we have also mentioned that first, our sequence trimming method accommodates for the trend that errors generally increase towards the end of the reads and second, no matter if errors from insertion, deletion or substitution are present, our de-biasing algorithm renders the sequences random.

The added clarification in the respective paragraph now reads as follows:

Other studies have shown that errors may be related to the sequencing context, where the highest rate of errors was found for nucleotide T, which in our case would result to be erroneous once in every 1,250 nucleotides, and the error rates for C, G, and A being much lower. Overall, expected error rates were found to increase towards the end of the DNA strand. To minimize the influence of (especially) deletion errors on randomness, we shortened all sequences to 60 nucleotides and simultaneously selected only the sequences containing the correct length of random nucleotides (Supplementary Information 1).” (Pfeiffer, F. *et al.* Systematic evaluation of error rates and causes in short samples in next-generation sequencing. *Sci. Rep.* **8**, 1–14 (2018). and Meiser, L. C. *et al.* Reading and writing digital data in DNA. *Nat. Protoc.* **15**, 86–101 (2020). and Schirmer, M., D’Amore, R., Ijaz, U. Z., Hall, N. & Quince, C. Illumina error profiles: Resolving fine-scale variation in metagenomic sequencing data. *BMC Bioinformatics* **17**, 1–15 (2016).)

In Figure 4 caption, the authors seem to be switching between the terms “primer” and “adapter”. This should be clarified.

We have adjusted the caption such that the word “adapter” is uniformly used. (This figure is no longer part of the main text, but can now be found in Supporting Information 1):

“Figure S1a: DNA Processing. Due to synthesis and sequencing errors, DNA strands may be shorter than 64 nucleotides. Thus, when reading 64 nucleotides, part of the adapter region (which is constant and not random) could unintentionally be read. This induces additional bias into the data. The following procedure has thus been adapted to ensure the absence of adapter regions in data: 1) Find all sequences containing the adapter and include these sequences in the new pool. 2) Shorten these sequences to 69 nucleotides. 3) Find all sequences still containing the first nine nucleotides of the adapter and exclude those from the pool. 4) Shorten all sequences left in the pool to 60 nucleotides.”

In Page 12, the bottom paragraph needs to be rephrased.

We thank the reviewer for this suggestion and have rephrased the respective paragraph to the following, in order to clarify the summary of results:

It was observed that the cumulative sum before de-biasing was skewed (binomial distribution not centered around a cumulative sum across of zero). Removing the bias by applying the Von Neumann algorithm shows a shift of the binomial distribution along the horizontal axis, such that when de-biased, the cumulative sum of the bit distribution is centered around zero. The effect of de-biasing on the bits can also be quantified as follows: Synthesis 1 by Microsynth with a nucleotide-to-bit mapping $A \rightarrow 0$, $C \rightarrow 0$, $T \rightarrow 1$, $G \rightarrow 1$, results in a de-biasing efficiency of 23.7% (meaning 23.7% of bits originally present in raw bit streams are still present after Von Neumann de-biasing). Although the loss of data is massive (more than 75% of all bits lost) and computational efficiency is low (as the average output rate of data is four times slower than the average input rate of data), bias removal is perfect (with the output being completely unbiased, as seen when comparing the cumulative sum across bit streams before and after de-biasing in Figure 4). (Dichtl, M. *Bad and good ways of post-processing biased physical random numbers*. (Springer, 2007).)

In page 12, the terms “computational efficiency” and “perfect bias removal” should be defined and the claims supported.

We have made the following modifications for added clarification and support in the text:

“Although the loss of data is massive (more than 75% of all bits lost) and computational efficiency is low (as the average output rate of data is four times slower than the average input rate of data), bias removal is perfect (with the output being completely unbiased, as seen when comparing the cumulative sum across bit streams before and after de-biasing in Figure 4).” (Dichtl, M. *Bad and good ways of post-processing biased physical random numbers*. vol. 4593 (Springer, 2007).)

Figure 4 illustrates that bias removal is perfect, thus supporting this claim. (The cumulative sum across bit streams is centered around 0 for de-biased bits).

In Table 1, the caption sentence should be corrected. “NIST statistical tests performed on processed synthesized DNA oligonucleotides ...”

We thank the reviewer and have updated the caption as suggested to

“NIST statistical tests performed on processed synthesized DNA oligonucleotides, mapped to bits using the scheme $A \rightarrow 0$, $C \rightarrow 0$, $T \rightarrow 1$, $G \rightarrow 1$. For each test, 56 bit streams containing 1096 bits were tested.”

In Figure 6, panel c is the bit distribution after de-biasing. It is not clear how the bit stream length is still 60 bits after de-biasing. Is this due to some sort of splitting of the larger concatenated bit blocks? This should be clarified.

We have added a sentence to the figure caption for clarification:

“Subsequently to Von Neumann de-biasing, the individual bit streams were combined into a block of bits and then separated into 60-bit streams for better comparison.”

We have additionally appended the supporting information with an explanatory illustration, shown as Supplementary Information 7, in order to clarify bit stream analysis procedures. The illustration looks as follows:

Similarly, in the Sup material regarding the 2-bits-1-nucleotide mapping should contain 120 bits per raw bit stream but it shows only 60 bits.

We thank the reviewer for having noticed this and have of course adjusted the plots as suggested. The figure in Supplementary Information 5 now shows the bit streams of length 120.

There are multiple erroneous and unclear references. See for example R2, R27-29. Please revise the references. This is a major presentation flaw

We thank the reviewer for this input. We have gone through all references and made sure that they adhere to Nature style of citation.

Reviewers' Comments:

Reviewer #1:

Remarks to the Author:

The authors have adequately addressed all of my concerns, and I support publication.

Reviewer #2:

Remarks to the Author:

We accept all the changes as addressing the first round of reviews, including the change in the paper structure.

Some typos still persist, e.g:

Synthesis of Nucleoc Acids

Response to reviewers

Reviewer #1:

The authors have adequately addressed all of my concerns, and I support publication.

We would like to thank the referee for supporting our work.

Reviewer #2:

We accept all the changes as addressing the first round of reviews, including the change in the paper structure.

Some typos still persist, e.g:

Synthesis of Nucleoc Acids

We would like to thank the referee for his/her support and have corrected the typos.